# The Relationship between Competitive Class Climate and Cyberloafing among Chinese Adolescents: A Curvilinear Moderated Mediation Model

**DOI:** 10.3390/ijerph20064705

**Published:** 2023-03-07

**Authors:** Shun Peng, Xiuhan Huang, Lei Xu, Shuangshuang Cai, Jiwen Chen, Hua Dong

**Affiliations:** Department of Psychology, School of Education, Jianghan University, Wuhan 430014, China

**Keywords:** competitive class climate, cyberloafing, perceived stress, self-esteem, curvilinear moderated mediation model

## Abstract

Since COVID-19 was officially listed as a pandemic, online schooling has become a more pervasive form of learning, and cyberloafing has become a widespread behavior, even among adolescents. However, less research has explored the influencing mechanism of adolescents’ cyberloafing. Based on relevant studies and the real lives of adolescents, this study aimed to examine the association between a competitive class climate and cyberloafing among adolescents, its underlying mechanism, the mediating role of perceived stress and the moderating role of self-esteem. A total of 686 adolescents were recruited to complete a set of questionnaires assessing cyberloafing, perceived stress, self-esteem, and perceived competitive class climate. The results showed that a competitive class climate was positively associated with perceived stress, and the U-shaped relationship between perceived stress and cyberloafing was significant. Perceived stress mediated the relationship between a competitive class climate and cyberloafing. Meanwhile, self-esteem moderated the U-shaped relationship between perceived stress and cyberloafing and the linear relationship between a competitive class climate and perceived stress. The results of this study indicate that the influence of a competitive class climate on individual learning behavior may be nonlinear, and proper competition can contribute to reducing individual cyberloafing.

## 1. Introduction

New coronavirus disease 2019 (COVID-19) is caused by a novel coronavirus (SARS-CoV-2) that is transmitted through aerosols and droplets [1]. In March 2020, it was officially listed as a global pandemic [2]. To reduce the spread of the virus and the likelihood of becoming infected, most countries recommend that people adopt or partially practice health behaviors such as social distancing [2]. In addition, online schooling has become a prevalent option, as classrooms are too intensive in terms of human contact. From the educational perspective, the academic use of internet technology allows students to be exposed to more timely, relevant, and up-to-date materials to realize positive outcomes from access to the internet [3]. However, researchers have found that, when laptops are allowed in classes, students usually send and receive emails, browse news and sports, download music, chat, play online games, browse blogs, visit social networks, and update personal homepages [4,5,6,7,8]. According to another survey, 70% to 90% of students sent an average of 12 text messages per class [9]. Students’ use of the internet in the classroom to have fun online rather than for academic purposes has an impact on their learning [10]. Researchers have suggested that engaging in non-study online behaviors while studying can be termed cyberloafing [11]. Cyberloafing refers to an individual’s spontaneous use of the internet to browse websites unrelated to work/study and to send and receive emails or text messages unrelated to work/study while working/studying [12]. However, cyberloafing has primarily been studied in the context of workplaces [13]. The development of wireless networks and mobile devices in educational settings has begun to draw widespread attention from educational researchers [11,14,15,16]. Empirical studies have suggested that cyberloafing may reduce one’s work or learning efficiency [17,18] and it has a negative impact on class performance, such as degrading the motivation of classmates and teachers, causing distraction, distorting the courses [19], and even having a devastating effect on students’ academic performance and physical and mental wellbeing [11,20]. Given this, exploring its influencing factors in an educational context has important practical implications for teaching implementation and classroom management.

### 1.1. The Relationship between a Competitive Class Climate and Adolescents’ Cyberloafing

According to Bronfenbrenner’s ecological systems theory [21], the environment has a direct and profound impact on individuals’ minds and behaviors. Classrooms, where students learn in groups, mold students’ behaviors and cognitive abilities [22,23]. Therefore, the classroom climate can directly affect students’ emotions and behaviors [24,25]. In East Asia, families and schools expect students to perform better than their peers and thus achieve higher academic success [26]. As a result, the classroom climate in China tends to be competitive. A competitive class climate refers to the psychosocial environment resulting from competition among classmates in academic and other areas [27]. Although there are no current studies directly exploring the relationship between a competitive class climate and cyberloafing, the existing studies do provide us with indirect evidence.

Researchers have focused on the complex effects of a competitive climate on job performance, such as David, Kim, Rodgers, and Chen [28], who showed that a competitive climate has an inverted U-shaped relationship with job performance. In other words, as the level of the competitive climate increases, individuals’ work engagement and work performance levels also increase, thus decreasing their non-work behaviors (e.g., cyberloafing), but when the competitive climate reaches a certain level, individuals’ work engagement and work performance begin to decrease, thus increasing their non-work activities (e.g., cyberloafing). Additionally, the Yerkes–Dodson law posits that the relationship between motivation and productivity is not linear, but rather takes an inverted U-shape [29]. If the drive to learn is too intense, it can create anxiety and tension, which interfere with the smooth flow of memory and thinking activities and make learning less effective. Thus, we speculate that, when adolescents have a less competitive class climate (extrinsic motivation), they may have more opportunities to engage in cyberloafing because they do not have to account for competition, and when adolescents have a more competitive class climate, they need more time to deal with competition and are thus unlikely to engage in cyberloafing. When adolescents have an excessively competitive class climate, they do not have enough time to cope with the situation or fall under great stress, and their level of cyberloafing increases again. Therefore, the present study proposes the hypothesis that a competitive class climate and cyberloafing have a U-shaped relationship (H1).

### 1.2. The Mediating Role of Perceived Stress

Perceived stress is the stress an individual experiences when he or she copes with stressful events in life [30]. An individual’s perceived stress can be affected by many factors, such as personal characteristics and environmental variables (e.g., family, school, and classroom) [31,32]. Previous studies have shown that a competitive environment can lead to uncertainty and feeling threatened [33], and uncertainty and feelings of threat from a competitive situation likely lead to perceived stress [34]. According to the process–person–context–time model [35], the external environment can affect the psychological states and behaviors of individuals [36]. When individuals are in a competitive class climate, to win, they will constantly push themselves, generating higher stress. Empirical studies have also validated this view, such as those of Fletcher, Major, and Davis [33], which showed that a competitive psychological climate was associated with greater stress.

Previous studies have suggested a significant positive correlation between perceived stress and cyberloafing; that is, people with high levels of perceived stress tend to engage in cyberloafing [15,37]. However, some researchers have argued that a lower level of perceived stress could also lead to cyberloafing in an attempt to cope with boredom [13]. Thus, people with medium levels of perceived stress have fewer cyberloafing behaviors and higher learning and work efficiency [13,15], while those who have too high or too low levels of perceived stress may engage in cyberloafing. Excessive perceived stress may lead to cyberloafing because it consumes an individual’s cognitive resources and physical and mental energy, leading to withdrawal behavior (e.g., cyberloafing) [4,38]. A low level of perceived stress may also cause people’s cyberloafing behaviors because they may feel that learning or work tasks are easy to complete or boring; thus, they have enough time to do them or do not want to do them [13]. Following this logic, we hypothesize that perceived stress and cyberloafing have a U-shaped relationship, i.e., as the level of perceived stress increases, individuals’ cyberloafing may gradually decrease. However, when the perceived stress level continues to increase, individuals may be overwhelmed; thus, they again exhibit cyberloafing (H2).

In addition, the process–person–context–time model suggests that there is a “distal and proximal” relationship among the factors that influence individual development. For individual behavior, environmental factors are often considered to be distal factors, while individual cognitive and psychological responses are considered to be proximal factors [39]. Distal factors can influence individual behavior through the mediation of proximal factors [35]. In this study, a competitive classroom climate can be considered as a distal factor that affects cyberloafing through perceived stress (proximal factor) [33,40]. Therefore, this study advances the hypothesis that perceived stress mediates the relationship between a competitive class climate and cyberloafing (H3).

### 1.3. The Moderating Role of Self-Esteem

Self-esteem, as an important trait of the self-system, is closely related to individuals’ social adaptability [41]. Researchers claim that self-esteem can act as a buffer against negative daily life events or stressors [42,43]. According to the anxiety-buffer hypothesis, self-esteem buffers the negative effects of stress on individuals. When individuals’ self-esteem levels are high, they can better cope with stressful events, thereby alleviating the negative impact of stress on themselves, while low-self-esteem individuals lack sufficient resources to deal with stress, which leads to negative consequences [44]. Studies have found that the positive relationship between stress and prenatal depression is weaker in individuals with high self-esteem than in individuals with low self-esteem [45]. The negative consequences of stress may be weaker in individuals with high self-esteem owing to the buffering effect of self-esteem on the negative effects of stress.

Additionally, research has shown that self-esteem is positively related to self-control. For example, low self-esteem leads to a loss of self-control [46,47]. People with low self-esteem may face substantial external stress; for example, they may have problems dealing with interpersonal relationships and coping with life events. As a result, they indulge in virtual networks in the hopes of escaping from the real world’s troubles and obtain a sense of success, belongingness, and recognition and understanding from others on the internet [48]. The internet has become an important platform for people with low self-esteem to express their emotions, release stress, and gain online psychological satisfaction [49]. Based on previous studies and theories, this study proposed the following hypothesis: the relationship between a competitive class climate and perceived stress is moderated by self-esteem. Specifically, the relationship between a competitive class climate and perceived stress is weaker among high-self-esteem individuals than among low-self-esteem individuals (H4). The U-shaped relationship between perceived stress and cyberloafing is moderated by self-esteem. Specifically, high self-esteem reduces cyberloafing whenever the perceived stress is too high or too low (H5). Self-esteem moderated the relationship between competitive class climate and cyberloafing, with a more robust U-shaped relationship between competitive class climate and cyberloafing seen clearly in individuals with high self-esteem compared with those with low self-esteem (H6). We demonstrated the hypothesized model among the four variables by constructing a curvilinear moderated mediation model (see Figure 1).

## 2. Methods

### 2.1. Participants and Procedures

The present study was approved by the Scientific Research Ethics Committee of the university. As it is difficult to conduct large-scale face-to-face questionnaire surveys during the COVID-19 pandemic (March 2021), this study conducted an online survey through WeChat to test two middle schools and three high schools in Hunan Province, China. After obtaining the participants’ informed consent, the graduate assistant administered the measures. They used a standard set of instructions to explain the requirements for these measures to all participants in the student class group (Wechat group), emphasizing the importance of the authenticity and integrity of their answers and assuring them that their answers would be kept confidential. A total of 700 adolescents participated in this study, among which 9 failed to sign the informed consent form and 5 failed to pass the trap questions. Finally, 686 valid data points were retained, with an effective rate of 98.000%. Among the retained participants, 38.192% were male and the rest were female (61.808%). The average age of the participants was 15.353 (range = 13~18, *SD* = 1.376).

### 2.2. Measures

#### 2.2.1. Competitive Class Climate

The competitive class climate was measured by the competitive subscale of the My Class Scale (MCS) [27]. The subscale includes seven items (e.g., “Everyone wants to be better than everyone else”). The responses are measured using a five-point Likert scale (from “0 = never” to “4 = always”). The higher the score, the more competitive the class climate. In the present study, Cronbach’s α of the subscale was 0.707.

#### 2.2.2. Cyberloafing

The Cyberloafing Scale used in this study was compiled by Blau et al. [50]. It contains a total of 16 items (e.g., “Browse general news websites”) with responses measured using a four-point Likert scale (from “1 = hardly ever/once every few months or less” to “4 = frequently/at least once a day”) [50]. During the test, the subjects were asked to answer questions about their real-life cyberloafing. Higher scores indicate more cyberloafing. The Cronbach’s α of the scale was 0.788. Validation factor analysis showed that the scale had good construct validity in this study (χ^2^/*df* = 4.086, RMSEA = 0.067, CFI = 0.945, TLI = 0.917, SRMR = 0.036).

#### 2.2.3. Perceived Stress

The Perceived Stress Scale-14 (PPS-14) used in this study was compiled by Cohen et al. [30] and consists of 14 items (e.g., “In the last month, how often have you felt that you were on top of things?”) with responses measured using a five-point Likert scale (from “0 = never” to “4 = very often”). Higher scores indicate greater pressure perceived by the individual. The Cronbach’s α in this study was 0.715. Validation factor analysis showed that the scale had good construct validity in this study (χ^2^/*df* = 3.369, RMSEA = 0.063, CFI = 0.953, TLI = 0.931, SRMR = 0.060).

#### 2.2.4. Self-Esteem

We used the Rosenberg Self-Esteem Scale (RSES) to measure adolescents’ self-esteem. The RSES contains two subscales: a positive self-esteem subscale and a negative self-esteem subscale [51,52]. Among them, individuals with high positive self-esteem have higher life satisfaction and tend to affirm their advantages and abilities, while negative self-esteem scores are more related to psychological and emotional disorders, and individuals with high scores tend to doubt their value and effectiveness [53]. In this study, we are interested in understanding the relationship between perceived stress and cyberloafing under high or low levels of positive self-esteem. Therefore, we used only the positive self-esteem subscale of the RSES to assess self-esteem [54]. The subscale includes five items (e.g., “On the whole, I am satisfied with myself”) with responses measured using a four-point Likert scale (from “1 = strongly disagree” to “4 = strongly agree”). The Cronbach’s α of the subscale in the present study was 0.737.

### 2.3. Data Analysis Method

First, we conducted descriptive statistical analysis using the psych package [55] in R [56]. Second, we followed the suggestion of Hayes et al. [57]; the indirect effect in the curvilinear mediation model can be denoted as θ and estimated as the product of the first partial derivative of the function of mediation variable (M) with respect to the independent variable (*X*) and the first partial derivative of the function of the dependent variable (*Y*) with respect to the mediation variable (*X*). The lavaan package was used to test the hypotheses.
(1)$$M=r_{00}+r_{c0}\left(controls\right)+r_{11}X+e_{1}$$
(2)$$Y=r_{00}+r_{co}\left(controls\right)+r_{21}X+r_{22}X^{2}+r_{23}M+r_{24}M^{2}+e_{2}$$

According to
(3)$$\theta =\frac{\partial Me}{\partial X}\frac{\partial Y}{\partial Me}$$

We derived the partial derivative of perceived stress to cyberloafing from Equation (1) and the partial derivative of competitive class climate to perceived stress from Equations (1)–(3):(4)$$\frac{\partial Me}{\partial x}=r_{11}$$
(5)$$\frac{\partial Y}{\partial Me}=r_{23}+2r_{24}M$$

According to Equations (4) and (5), the instantaneous indirect effect of competitive class climate on cyberloafing through perceived stress is as follows:(6)$$\theta =r_{11}\left(r_{23}+2r_{24}M\right)$$

As our hypotheses involve mediated moderation with nonlinear effects, we propose the following steps to test our hypotheses.
(7)$$M=r_{00}+r_{co}\left(controls\right)+r_{31}X+r_{32}W+r_{33}XW+e_{3}$$
(8)$$Y=r_{00}+r_{co}\left(controls\right)+r_{41}X+r_{42}X^{2}+r_{43}M+r_{44}M^{2}+r_{45}MW+r_{46}M^{2}W+r_{47}XW+r_{48}X^{2}W+r_{49}W+e_{4}$$

We followed the procedures of Lind and Mehlum [58] to test the U-shaped relationship between perceived stress and cyberloafing. The $r_{30}$ in Equations (2) and (3) has to be positive and significant, and the confidence interval of the inflection point (−γ202γ30) is within the value range of the independent variable. We used Equations (1) and (2) to test the curvilinear mediation model and Equations (1) and (3) to test the curvilinear moderated mediation model. Additionally, $r_{60}$ indicates whether the U-shaped relationship between perceived stress and cyberloafing varies as a function of self-esteem.

Next, we derived the partial derivative of perceived stress to cyberloafing from Equation (1) and the partial derivative of competitive class climate to perceived stress from Equations (2) and (3):(9)$$\frac{\partial Me}{\partial x}=r_{31}+r_{33}W$$
(10)$$\frac{\partial Y}{\partial Me}=r_{43}+2r_{44}M+r_{45}W+2r_{46}MW$$

Combining Equations (4) and (7), the moderated instantaneous indirect effect of perceived stress is as follows:(11)$$\theta =\left(r_{31}+r_{33}W\right)\left(r_{43}+2r_{44}M+r_{45}W+2r_{46}MW\right)$$

In Equations (8) and (9), θ is not a constant but a linear function of the product term of perceived stress and self-esteem. Mathematically, if the difference in θ at low versus high levels of perceived stress and self-esteem is significantly different from zero, the curvilinear mediated moderation model is supported.

## 3. Results

### 3.1. Common Method Bias Test

As suggested by Podsakoff et al. [59], this study controlled for common method biases by collecting questionnaires anonymously during the administration process. Common method bias is an artificial covariation between predictor and criterion variables due to the use of the same data sources, the same measuring environment, the context of the items, or the characteristics of the items themselves. Such artificial covariation can seriously confuse the results and potentially mislead the conclusions. It is a kind of systematic error and thus needs to be controlled. Researchers often adopt Harman’s single-factor test to test common method bias for the sake of study rigor. The basic assumption of this method is that, if there is a large amount of variation in the common method, the factor analysis will separate out a single factor or a common factor that explains most of the variation. In this study, the explanation rate of all items on the first common factor was 15.374%, indicating that there was no serious common method bias in this study.

### 3.2. Descriptive Statistics and Correlation Coefficients

The means and standard deviations of age, competitive class climate, perceived stress, self-esteem, and cyberloafing, as well as the Pearson correlation coefficients among the last four variables, are shown in Table 1. We can see from Table 1 that competitive class climate was positively correlated with perceived stress, cyberloafing, and self-esteem. Perceived stress was negatively correlated with self-esteem and positively associated with cyberloafing. Self-esteem was negatively correlated with cyberloafing.

### 3.3. Testing the Mediating Role of Perceived Stress

First, we examined the U-shaped relationship between competitive class climate and cyberloafing, and the results showed that competitive class climate squared was positively correlated with cyberloafing (β = 0.271, *p* < 0.001), supporting Hypothesis 1. Second, we constructed two models to test Hypotheses 2–6. The results of the mediation model (see Table 2) show that competitive class climate had a positive relationship with perceived stress (β = 0.185, *p* < 0.001). The relationship between the quadratic of competitive class climate and cyberloafing was not significant (β = −0.001, *p* = 0.981). Perceived stress was positively associated with cyberloafing (β = 0.082, *p* = 0.007), and perceived stress squared was positively correlated with cyberloafing (β = 0.123, *p* < 0.001), thus supporting H2. To test the indirect effect of perceived stress, we used Hayes et al.’s [57] procedure to calculate *θ*. When perceived pressure was at a lower level (*M* − *SD*), *θ* was −0.029 (95% CI = [−0.056, −0.001]). When perceived stress levels were high (*M + SD*), *θ* was 0.059 (95% CI = [0.028, 0.090]). The difference between the two *θ* estimates was −0.088 (95% CI = [−0.140, −0.035]), indicating that the mediating effect of perceived stress (Hypothesis 3) was supported; perceived stress plays a fully mediating role between competitive class climate and cyberloafing. This also suggests that, despite the direct effect between competitive class climate and cyberloafing not being significant, competitive class climate can also impact cyberloafing through an indirect curve effect.

### 3.4. Testing the Moderating Role of Self-Esteem

We followed the suggestion of Lin et al. [60] to test the curvilinear moderated mediation model (see Table 3). The results indicated that competitive class climate was positively correlated with perceived stress (β = 0.186, *p* < 0.001). The interaction of competitive classroom climate and self-esteem was negatively associated with stress perception (β = −0.062, *p* = 0.005), and Hypothesis 4 was supported. The competitive class climate was positively associated with cyberloafing (β = 0.192, *p* < 0.001). The effect of competitive class climate squared and the interaction of competitive class climate squared with self-esteem on cyberloafing was not significant. Hypothesis 6 was not supported. Perceived stress (β = 0.105, *p* < 0.001) and perceived stress squared (β = 0.119, *p* < 0.001) were positively associated with cyberloafing. The interaction of self-esteem and perceived stress square was positively correlated with cyberloafing (β = 0.096, *p* = 0.001), thus supporting Hypothesis 5.

To provide a more explicit account of the moderating role of self-esteem in a competitive class climate on the relationship between perceived stress and cyberloafing, a simple effects analysis was conducted. The results showed that, at higher levels of self-esteem (*M + SD*), stress perception was high, and the effect value of stress perception on cyberloafing was 0.524 (*p* < 0.001). The effect value of stress perception on cyberloafing was −0.268 (*p* = 0.01) when stress perception was low, with an effective value difference of 0.792, 95% CI = [0.438, 1.146]; at lower levels of self-esteem (*M* − *SD*), the effect of stress perception on cyberloafing was 0.124 (*p* = 0.204) when stress perception was high and, to show the pattern of this interactive effect, we plotted the simple effects in Figure 2.

In addition, the positive predictive effect of competitive class climate on perceived stress was significant (β = 0.124, *p* < 0.001) when levels of self-esteem were high (*M* + 1*SD*); with lower levels of self-esteem (*M* − 1*SD*), the predictive effect of competitive class climate on perceived stress increased (β = 0.248, *p* < 0.001). We plotted the simple effects in Figure 3.

To further illustrate the moderating effect of self-esteem on the curvilinear indirect path, when self-esteem was high (*M + SD*), the difference in the indirect effect (θ) under different levels of perceived stress (*M ± SD*) was 0.099 (95% CI = [0.015, 0.182]). Similarly, when self-esteem was low (*M* − *SD*), the difference in the indirect effect θ under different levels of perceived stress (*M ± SD*) was 0.021 (95% CI = [−0.074, 0.116]).

## 4. Discussion

During COVID-19, online learning became the primary learning method for students worldwide, which provided fertile ground for cyberloafing. Our study focused on the impact of an individual live environment and competitive class climate on cyberloafing. Specifically, we explored the relationship between competitive class climate and cyberloafing with perceived stress as a mediator and self-esteem as a moderator. This study found a linear relationship between competitive class climate and perceived stress and a U-shaped relationship between perceived stress and cyberloafing. Perceived stress fully mediated the relationship between competitive class climate and cyberloafing, and self-esteem moderated this relationship.

### 4.1. The Mediating Role of Perceived Stress

The results of this study indicate that the relationship between competitive classroom climate and cyberloafing was fully mediated by perceived stress (Hypothesis 3 was supported). In the first path of the mediation process, we found that competitive class climate was positively associated with perceived stress. This means that, the more competitive a class is, the more stress the students perceive. This result is consistent with the findings of previous studies that showed that individuals’ environments can impact their psychology and behaviors [33,61,62].

In the second path of the mediation process, we found that the relationship between perceived stress and cyberloafing was U-shaped (Hypothesis 2 was supported). Specifically, the intensity of the students’ cyberloafing behaviors decreased as the level of their perceived stress increased until an inflection point. Then, their cyberloafing behaviors decreased with the continuous increase in the level of their perceived stress. This result is consistent with the findings of studies conducted by Graham et al. [63] and Wu et al. [11]. According to stress response theory, the stress response starts in the brain [64]. When individuals experience stressful events, such as a competitive class climate, their amygdala produces a “fight-or-flight” response [64,65]. When individuals perceive less stress, the stressful event is not perceived as a threat. Such a situation induces a “rest and digest” response. As perceived stress increases, we gradually experience threat. If the individual believes that the “threat” can be coped with, his or her amygdala produces a “fight” response. Thus, other unnecessary behaviors (e.g., cyberloafing) will be reduced. When the perceived stress reaches a certain level and the individual feels that he or she is unable to cope with the “threat”, his or her amygdala produces a “flight” response. This explanation shows that a moderate level of perceived stress is better, which is consistent with the basic idea of the threshold effect [64]. Thus, our results provide a new perspective on the relationship between perceived stress and cyberloafing.

This study found that the direct effect of competitive class climate and cyberloafing was not significant, thus Hypothesis 1 is not verified. On the one hand, this may be because of the fact that perceived stress might be closer to competitive class climate, and competitive climate may impact other variables through perceived stress. This is similar to the findings of Fletcher et al. [33], where a competitive psychological climate was associated with greater stress, but not directly related to self-rated task performance. On the other hand, the intensity of the competitive classroom climate in today’s society occurs within a “controlled range” and does not grow indefinitely, so it may not yet dampen other influences on adolescents’ intrinsic psychological traits [66]. The other factors that influence cyberloafing mainly affect adolescents’ levels of perceived stress, which in turn affect adolescents’ cyberloafing.

### 4.2. The Moderating Role of Self-Esteem

This study found that self-esteem moderates the relationship between competitive class climate and perceived stress; specifically, the relationship between competitive class climate and perceived stress is stronger in individuals with lower self-esteem than in those with higher self-esteem. Self-esteem is a person’s overall sense of self-worth or personal value [67,68]. Research has shown that individuals with high self-esteem tend to have high self-evaluations and confidence [69]. Therefore, they can better cope with stressful events (e.g., a competitive class climate), alleviate the negative impact of perceived stress on themselves, and maintain a proper level of stress. However, individuals with low self-esteem do not believe they can cope with problems and stress, leading to higher cyberloafing frequencies [44,70]. Therefore, when individuals perceive greater stress, self-esteem may serve as a buffer to mitigate the negative effects of perceived stress on cyberloafing.

We also found that self-esteem moderates the U-shaped relationship between perceived stress and cyberloafing. As the level of self-esteem increases, the U-shaped relationship between perceived stress and cyberloafing becomes less significant. This result is consistent with a previous study in which self-esteem served as a moderator of the relationship between environmental stimulation and risky behavior [71]. On the one hand, our results support the buffering hypothesis of self-esteem [44], which argues that self-esteem can buffer the negative effects of stress on individuals. When individuals have a high level of self-esteem, their self-confidence is high and they believe that they can cope with stressful events, which alleviates the negative impact of stress on themselves [72]. Conversely, individuals with low self-esteem have lower self-efficacy. They believe they cannot cope with stress and are more likely to avoid stressful events by engaging in off-task behaviors, such as cyberloafing [44,45]. Therefore, self-esteem can buffer individuals’ avoidance behaviors under stress. On the other hand, individuals with high self-esteem are usually goal-oriented and motivated in learning. They tend to make continuous efforts, are less influenced by the external environment, and are less likely to engage in behaviors unrelated to learning, even when they are under stress [73,74]. Compared with those with low self-esteem, the relationship between perceived stress and cyberloafing in individuals with high self-esteem is closer to a straight line. The moderating role of self-esteem in the U-shaped relationship between perceived stress and cyberloafing provides us with a comprehensive understanding of why students engage in cyberloafing. Self-esteem not only buffers the negative effects of stress, but also enables individuals to maintain their engagement in learning under low stress. Additionally, the present study did not find a moderating effect of self-esteem between competitive class climate and cyberloafing. This may be because of the fact that the effect of competitive class climate on cyberloafing occurs through perceived stress, and the direct relationship between them is not significant, which leads to a nonsignificant moderating effect of self-esteem between the two.

### 4.3. Limitations

The present study had several limitations. First, although our design was guided by established theories and previous findings, our data are cross-sectional. Investigations that include longitudinal data are needed to consolidate these relationships in future studies. Second, our study collected data from only a single source (e.g., self-report scale), and future studies should collect data from multiple sources (e.g., peer and teacher reports) to increase the objectivity of the study. Third, in our study, it was not clear whether individuals were cyberloafing to rest under high stress or as a go-slow behavior as a result of high stress. Future research should distinguish between the purposes of individual cyberloafing.

## 5. Conclusions

To the best of our knowledge, our study is the first to explore how and when the class climate affects cyberloafing. Our results showed that a competitive class climate increases students’ perceived stress in class and that perceived stress and cyberloafing have a U-shaped relationship. Moreover, self-esteem moderates both the U-shaped relationship between perceived stress and cyberloafing and the relationship between a competitive class climate and perceived stress. Our results have implications for both adolescents and teachers. For adolescents, our results suggest that individual self-esteem is effective in mitigating the negative consequences of individual stress, and adolescents can engage in simple reminiscence activities to increase their self-esteem levels in support groups [75]. For teachers, our results suggest that an appropriately competitive class climate is helpful in reducing individual cyberloafing. However, if the classroom climate is too competitive, it increases the risk of cyberloafing among adolescents; therefore, teachers need to construct student-centered course management programs that make classroom management more participatory, allow students to self-set standards of behavior, and build a reasonably competitive environment that encourages students to achieve multiple successes [76]. The results of this study suggest that the efforts of both teachers and adolescents can be effective in reducing cyberloafing among adolescents.

## Figures and Tables

**Figure 1 ijerph-20-04705-f001:**
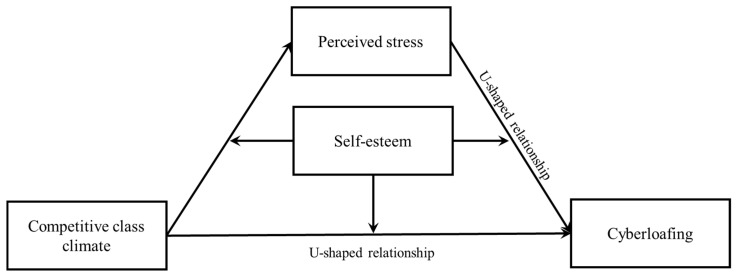
Hypothesized model.

**Figure 2 ijerph-20-04705-f002:**
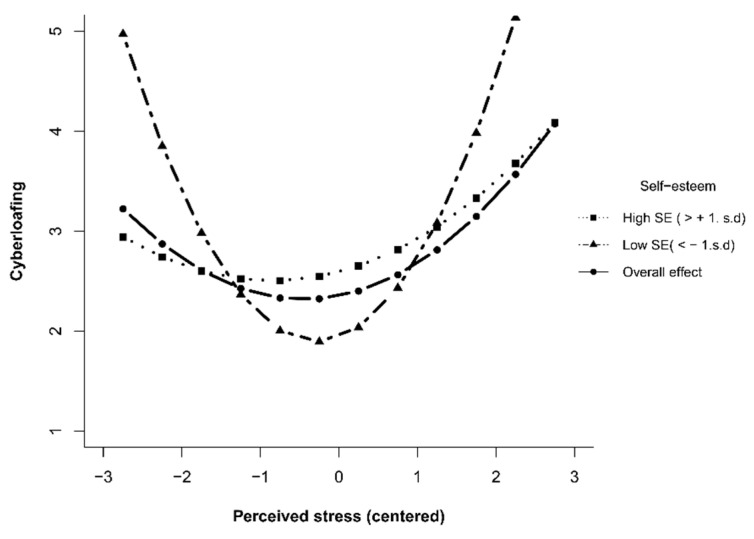
Relationship between perceived stress and cyberloafing as a function of self-esteem. Note. a. SE = self-esteem. b. Over effect means the relationship between perceived stress and cyberloafing in general.

**Figure 3 ijerph-20-04705-f003:**
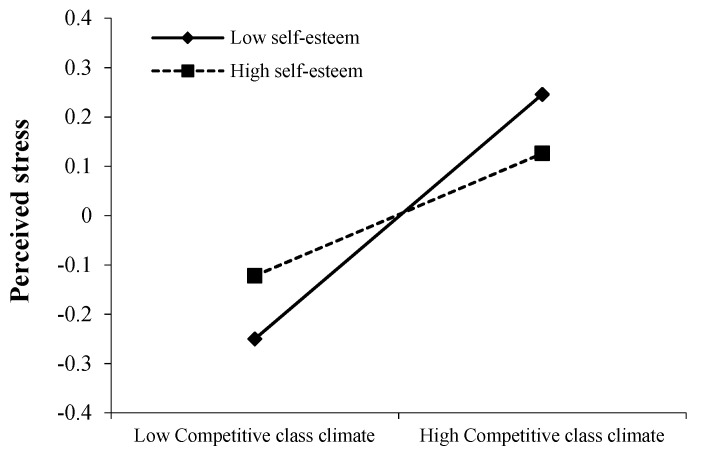
Relationship between perceived stress and competitive class climate as a function of self-esteem.

**Table 1 ijerph-20-04705-t001:** Mean, standard deviation, and correlation coefficients.

Variables	*M*	*SD*	1	2	3	4	5
1. Gender ^α^	—	—	—				
2. Age	15.353	1.376	—	—			
3. Competitive class climate	2.193	0.512	0.091 **	−0.013	—		
4. Perceived stress	2.925	0.452	0.150 *	−0.160 ***	0.210 ***	—	
5. Self-esteem	2.351	0.764	−0.087 *	0.076 *	−0.200 ***	−0.059	—
6. Cyberloafing	2.386	0.489	0.100 **	−0.026	0.280 ***	0.14 ***	−0.34 ***

Notes: *n* = 686. * *p* < 0.05, ** *p* < 0.01, *** *p* < 0.001. ^α^ 0 *=* male, 1 = female.

**Table 2 ijerph-20-04705-t002:** Test of the mediation model.

Variables	Dependent Variable: Perceived Stress			Dependent Variable: Cyberloafing		
	β	Boot *SE*	95%Boot CI	β	Boot *SE*	95%Boot CI
Age	−0.140 **	0.038	[−0.214, −0.066]	−0.007	0.032	[−0.070, 0.056]
Gender ^α^	0.123 **	0.036	[0.052, 0.194]	0.059	0.037	[−0.014, 0.131]
Competitive class climate (X)	0.185 ***	0.032	[0.123, 0.248]	0.260 ***	0.039	[0.183, 0.336]
X^2^				−0.001	0.035	[−0.070, 0.069]
Perceived stress (M)				0.082 **	0.034	[0.015, 0.148]
M^2^				0.123 ***	0.030	[0.063, 0.182]
	*R*^2^ = 0.078			*R*^2^ = 0.103		

Notes: *n* = 686. ** *p* < 0.01, *** *p* < 0.001. ^α^ 0 *=* male, 1 = female. Competitive class climate and perceived stress were centered on before calculating the second-order terms.

**Table 3 ijerph-20-04705-t003:** Test of the curvilinear moderated mediation model.

Variables	Dependent Variable: Perceived Stress			Dependent Variable: Cyberloafing		
	β	Boot *SE*	95% Boot CI	β	Boot *SE*	95% Boot CI
Age	−0.139 **	0.038	[−0.212, −0.065]	0.005	0.033	[−0.060, 0.071]
Gender ^α^	0.123 ***	0.036	[0.052, 0.194]	0.028	0.037	[−0.043, 0.100]
Competitive class climate (X)	0.186 ***	0.033	[0.121, 0.252]	0.192 ***	0.039	[0.115, 0.269]
Self-esteem (W)	0.002	0.038	[−0.072, 0.076]	−0.317 ***	0.046	[−0.407, −0.227]
W × X	−0.062 *	0.030	[−0.121, −0.002]	0.044	0.040	[−0.036, 0.123]
X^2^				0.015	0.037	[−0.057, 0.088]
W × X^2^				−0.034	0.056	[−0.144, 0.076]
Perceived stress(M)				0.105 ***	0.032	[0.042, 0.167]
M^2^				0.119 ***	0.030	[0.059, 0.178]
W × M				0.023	0.034	[−0.044, 0.090]
W × M^2^				−0.096 ***	0.039	[0.019, 0.172]
	*R*^2^ = 0.077			*R*^2^ = 0.181		

Notes: *n* = 686. * *p* < 0.05, ** *p* < 0.01, *** *p* < 0.001. ^α^ 0 *=* male, 1 = female. Competitive class climate and perceived stress were centered on before calculating the second-order terms.

## Data Availability

All data were available upon request.

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
