# Peer review of "The Relationship between Competitive Class Climate and Cyberloafing among Chinese Adolescents: A Curvilinear Moderated Mediation Model"

_ijerph, 2023, doi:10.3390/ijerph20064705_

Round 1
Reviewer 1 Report
First, i want to congratulated your research. However, i suggest you follow instructions for authors from this journal "References must be numbered in order of appearance in the text, In the text, reference numbers should be placed in square brackets [ ], and placed before the punctuation; for example [1], [1–3] or [1,3]."
Also, in the section 1.3, the figure 1 is confused due to the arrows does not have an order or sequence.
Finally, in the section conclusions could be improved it, because some sentences could be explained with major details due to your results have a great impact.
Author Response
- First, i want to congratulated your research. However, i suggest you follow instructions for authors from this journal "References must be numbered in order of appearance in the text, In the text, reference numbers should be placed in square brackets [ ], and placed before the punctuation; for example [1], [1–3] or [1,3]."
A: Thank you for your reminder. We have revised this in the manuscript.
- Also, in the section 1.3, the figure 1 is confused due to the arrows does not have an order or sequence.
A: Thank you for your reminder. We redrew the hypothesized model diagram to make the relationship between the variables more intuitive.
- Finally, in the section conclusions could be improved it, because some sentences could be explained with major details due to your results have a great impact.
A: Thank you for your reminder. Through reviewing the literature and the other reviewers' comments, we have added some specific suggestions to the manuscript. For adolescents, our results suggested that individual self-esteem is effective in mitigating the negative consequences of individual stress, and adolescents can seek support groups and engage in simple reminiscence activities to increase their self-esteem levels (Niveau, New, & Beaudoin, 2021). For teachers, our results suggest that an appropriate competitive class climate is helpful in reducing individual cyberloafing. However, if the classroom climate is too competitive, it can increase the risk of cyberloafing among adolescents; therefore, teachers need to construct student-centered course management programs that make classroom management more participatory, allow students to self-set standards of behavior, and build a reasonably competitive environment that encourages students to achieve multiple successes (Freiberg & Lamb, 2009). The results of this study suggest that the efforts of both teachers and adolescents can be effective in reducing cyberloafing among adolescents.
Freiberg, H. J., & Lamb, S. M. Dimensions of person-centered classroom management. Theory into practice. 2009, 48(2), 99-105. https://doi.org/10.1080/00405840902776228
Niveau, N., New, B., & Beaudoin, M. Self-esteem interventions in adults–a systematic review and meta-analysis. Journal of Research in Personality. 2021, 94, 104131. https://doi.org/10.1016/j.jrp.2021.104131
Reviewer 2 Report
Thank you for the opportunity to review this manuscript titled “The relationship between competitive class climate and cyber- loafing among Chinese adolescents: A curvilinear moderated mediation model”. The topic of the study was really interesting and important and has practical implications. The manuscript aroused some concerns that should be addressed before it can be accepted.
1. Introduction: (1) It seems that the relationship between competitive class climate and adolescents’ cyberloafing lack of direct research evidence, which makes the argument less convincing. In addition, the authors need to pay more attention to the definition of the concept, i.e., cyberloafing and competitive class climate. (2) The U-shaped relationship between perceived stress and cyberloafing should be mentioned in 1.2. The mediating role of perceived stress. Besides, “the following hypothesis was proposed in this study: a competitive class climate increases adolescents’ perceived stress.” is a hypothesis?The mediating role of perceived stress needs to be revised to further clarify ideas and increase logic.
2. Methods: (1) When (I mean the Year and month) was the survey conducted, as the authors mentioned the pandemic. Whether the survey is conducted online or offline? If it is conducted offline, in the classroom or in what situation? (2) The author should demonstrate the applicability of measurements in China, such as the Cyberloafing Scale, and the Perceived Stress Scale-14.
3. Results: The authors hypothesized that competitive class climate and cyberloafing have a U- shaped relationship (H1), the results showed that the relationship between the quadratic of competitive class climate and cyberloafing was not significant, but the authors said that the hypothesis 1 was supported. It should be explained how Hypothesis 1 was verified and from which results? (2) It should be noted that it is a complete mediation model.
4.Discussion: (1) I suggest that the authors should provide some specific suggestions to improve the class environment during online schooling, reduce cyberloafing and perceived stress and enhance self-esteem. (2) The direct effect of competitive class climate and cyberloafing was not significant, and H5 was not supported, the two results need to be discussed.
Author Response
Thank you for the opportunity to review this manuscript titled “The relationship between competitive class climate and cyber- loafing among Chinese adolescents: A curvilinear moderated mediation model”. The topic of the study was really interesting and important and has practical implications. The manuscript aroused some concerns that should be addressed before it can be accepted.
- Introduction: (1) It seems that the relationship between competitive class climate and adolescents’ cyberloafing lack of direct research evidence, which makes the argument less convincing. In addition, the authors need to pay more attention to the definition of the concept, i.e., cyberloafing and competitive class climate.
A: Thank you for your reminder. As the reviewer suggests, direct evidence for the current relationship between competitive class climate and cyberloafing is lacking, but available studies provide indirect evidence. Researchers have focused on the complex effects of competitive climate on individual job performance, such as David, Kim, Rodgers, and Chen (2021), who showed that competitive climate has an inverted U-shaped relationship with job performance; in other words, with increasing levels of competitive climate, the level of individual work engagement and performance also increase. When the competitive climate reaches a certain level, however, individual work engagement and work performance begin to decrease, thus increasing non-work behaviors (e.g., cyberloafing). Additionally, the Yerkes-Dodson law posits that the relationship between motivation and productivity is not linear but rather takes an inverted U-shape (Corbett, 2015). If the drive to learn is too intense, it can create anxiety and tension, which interfere with the smooth flow of memory and thinking activities and makes learning less effective. Therefore, we speculate that when adolescents have a less competitive class climate (extrinsic motivation), they have more opportunities to engage in cyberloafing because they do not have to account for competition, and when adolescents have a more competitive class climate, they need more time to deal with competition and are more likely to engage in cyberloafing. When adolescents have a sufficiently competitive class climate, they do not have enough time to cope with the situation or fall under great stress, and their level of cyberloafing increases again. Therefore, the present study proposes the hypothesis that competitive class climate and cyberloafing have a U-shaped relationship (H1).
In addition, regarding the definition of cyberloafing, Kamin (1995) first used the term to refer to the work deviant behavior of employees who surf the internet for entertainment during work hours (Kamin, 1995; Quinion, 2005). Later, different researchers named and defined this type of behavior differently. For example, Guthrie and Gray (1996) called it junk computing, which is the use of information systems in a way that is inconsistent with organizational goals, emphasizing its uselessness. Marron (2000) defined cyberloafing in terms of its harmfulness as: "The behavior of employees who waste time and company resources by engaging in online entertainment during working hours". Mills and Hu ( 2001) define cyberloafing as the behavior of employees who use the internet for entertainment or personal purposes at work. Lim ( 2002) considers cyberloafing to be the behavior of employees who spontaneously use the company network to browse non-work related websites and send and receive non-work related emails during work hours. Lim (2002)'s definition of cyberloafing has been widely cited and argued; therefore, we apply his view to the learning environment and define cyberloafing as referring to an individual's spontaneous use of the internet to browse websites unrelated to work/study and sending and receive emails or text messages unrelated to work/study while working/studying.
According to Jiang (2004), competitive class climate refers to the psychosocial environment resulting from competition among classmates in academic and other areas (Jiang, 2004). We have revised the manuscript according to the reviewers' comments.
Corbett, M. From law to folklore: work stress and the Yerkes-Dodson Law. Journal of Managerial Psychology. 2015, 30(6): 741-752. https://doi.org/10.1108/jmp-03-2013-0085
David E M, Kim T Y, Rodgers M, et al. Helping while competing? The complex effects of competitive climates on the prosocial identity and performance relationship. Journal of Management Studies. 2021, 58(6): 1507-1531.
Jiang G.R. Classroom environments in primary and secondary schools: structure and measurement. Psychological Science. 2004, 27(4), 839-843. https://doi.org/10.16719/j.cnki.1671-6981.2004.04.018
Kamin L J. Behind the curve. Scientific American, 1995, 272(2), 99-103.
Lim, V. K. The IT way of loafing on the job: Cyberloafing, neutralizing and organizational justice. Journal of organizational behavior: the international journal of industrial, occupational and Organizational Psychology and Behavior. 2002, 23(5), 675-694. https://doi.org/10.1002/job.161
Marron,K. Attack of the cyberslackers. The ( London) Globe and Mail. 2000,January 20,p. T5.
Mills J E, Hu B, Beldona S, et al. Cyberslacking! A liability issue for wired workplaces. Cornell Hotel and Restaurant Administration Quarterly. 2001, 42(5), 34-47. https://doi.org/10.1016/S0010-8804(01)80056-2
(2) The U-shaped relationship between perceived stress and cyberloafing should be mentioned in 1.2. The mediating role of perceived stress. Besides, “the following hypothesis was proposed in this study: a competitive class climate increases adolescents’ perceived stress.” is a hypothesis?The mediating role of perceived stress needs to be revised to further clarify ideas and increase logic.
A: Thank you for this reminder. According to reviewers’ comments, we supposed that perceived stress and cyberloafing have a U-shaped relationship; Specifically, as the level of perceived stress increases, individuals’ cyberloafing activities tend to decrease. However, when the perceived stress level continues to increase, individuals may become overwhelmed and again resort to cyberloafing (H2). In addition, relevant studies have confirmed that competitive psychological climate is associated with greater levels of stress (Fletcher, Major, & Davis, 2008).Therefore, the following hypothesis that had been proposed in this study, namely, “a competitive class climate increases adolescents’ perceived stress” is indeed not a hypothesis. This is a writing issue, and we have revised the manuscript to account for it.
In accordance with the reviewer's comment, we reinforce the hypothesis regarding the mediating effect of perceived stress. In addition, The Process-Person-Context-Time Model suggests that there is a “distal and proximal” relationship among the factors that influences individual development. For individual behavior, environmental factors are often considered to be distal factors, while individual cognitive and psychological responses are considered to be proximal factors (Neblett & Cortina, 2006). Distal factors can influence individual behavior through the mediation of proximal factors (Bronfenbrenner, 2005). In this study, competitive classroom climate can be considered as a distal factor that affects cyberloafing through perceived stress (proximal factor) (Fletcher, Major, & Davis, 2008; Zhou, Li, Hai, Wang, & Niu, 2021). Therefore, this study hypothesizes that perceived stress mediates the relationship between competitive class climate and cyberloafing (H3).
Bronfenbrenner, U. Making human beings human: Bioecological perspectives on human development. 2005, sage.
Fletcher, T. D., Major, D. A., & Davis, D. D. The interactive relationship of competitive climate and trait competitiveness with workplace attitudes, stress, and performance. Journal of Organizational Behavior: The International Journal of Industrial, Occupational and Organizational Psychology and Behavior. 2008,29(7), 899-922. https://doi.org/10.1002/job.503
Neblett, N. G., & Cortina, K. S. Adolescents’ thoughts about parents’ jobs and their importance for adolescents’ future orientation. Journal of Adolescence. 2006 ,29(5), 795-811. https://doi.org/10.1016/j.adolescence.2005.11.006
Zhou, B., Li, Y., Hai, M., Wang, W., & Niu, B. Challenge-hindrance stressors and cyberloafing: A perspective of resource conservation versus resource acquisition. Current Psychology. 2021,42, 1-10. https://doi.org/10.1007/s12144-021-01505-0
- Methods: (1) When (I mean the Year and month) was the survey conducted, as the authors mentioned the pandemic. Whether the survey is conducted online or offline? If it is conducted offline, in the classroom or in what situation?
A:The data for this study were obtained through an online survey administered during the COVID-19 pandemic (March, 2021). The main process of its administration occurred after the teacher finished the lesson, and it involved the research assistant explaining the instructional phrase to the students and then presenting the questionnaire QR code, which the students scanned to access the survey, and responded to digitally.
(2) The author should demonstrate the applicability of measurements in China, such as the Cyberloafing Scale, and the Perceived Stress Scale-14.
A: Thank you for this reminder. First, this study found that the PSS-14 and cyberloafing scale had good reliability in the sample we drew. Second, the validation factor analysis showed that the PSS-14 had good construct validity for this scale in this study (χ2/df = 3.369, RMSEA=0.063, CFI=0.953, TLI=0.931, SRMR=0.060). The cyberloafing scale also had good construct validity for this scale in this study (χ2/df = 4.086, RMSEA=0.067, CFI=0.945, TLI=0.917, SRMR=0.036). These results all indicate that the cyberloafing scale and PSS-14 have good reliability and validity in the sample selected for this study.
- Results: The authors hypothesized that competitive class climate and cyberloafing have a U- shaped relationship (H1), the results showed that the relationship between the quadratic of competitive class climate and cyberloafing was not significant, but the authors said that the hypothesis 1 was supported. It should be explained how Hypothesis 1 was verified and from which results? (2) It should be noted that it is a complete mediation model.
A: As suggested by the reviewer, this study did not find a direct U-shaped relationship between competitive class climate and cyberloafing, but this study found that the indirect effect of competitive class climate on cyberloafing through perceived stress (which does have a U-shaped relationship) was significant. Therefore, we concluded that a U-shaped relationship between competitive class climate and cyberloafing still exists. In calculating only the relationship between competitive class climate and cyberloafing, we found that competitive class climate squared was positive with cyberloafing (β = 0.271, p < 0.001).
In addition, Preacher and Hayes (2008) call for abandoning the concept of "full mediation" and treating all mediators as partial mediators. Due to this controversy, we had not specified the full mediation model in the manuscript. We have revised the manuscript according to the reviewer's comment.
Hayes, A. F., & Preacher, K. J. Quantifying and testing indirect effects in simple mediation models when the constituent paths are nonlinear. Multivariate behavioral research. 2010, 45(4), 627-660. https://doi.org/10.1080/00273171.2010.498290
- Discussion:(1) I suggest that the authors should provide some specific suggestions to improve the class environment during online schooling, reduce cyberloafing and perceived stress and enhance self-esteem.
A: In conjunction with other reviewers' comments, we included in the conclusion the idea that for adolescents, our results suggest that individual self-esteem is effective in mitigating the negative consequences of individual stress and that adolescents can seek support groups and engage in simple reminiscence activities to improve their self-esteem levels (Niveau, New, & Beaudoin, 2021). For teachers, our results suggest that an appropriately competitive class climate is helpful in reducing individual cyberloafing, but it is also important to note that if the classroom climate is too competitive, it can increase the risk of adolescent cyberloafing, therefore, teachers need to construct student-centered lesson management programs that make classroom management more participatory, allow students to set their own standards of behavior, and build a reasonably competitive environment that encourages students to achieve multiple successes (Freiberg & Lamb, 2009). The results of this study suggest that the efforts of both teachers and youth can be effective in reducing youth cyberloafing.
Freiberg, H. J., & Lamb, S. M. Dimensions of person-centered classroom management. Theory into practice. 2009, 48(2), 99-105. https://doi.org/10.1080/00405840902776228
Niveau, N., New, B., & Beaudoin, M. Self-esteem interventions in adults–a systematic review and meta-analysis. Journal of Research in Personality. 2021, 94, 104131. https://doi.org/10.1016/j.jrp.2021.104131
(2) The direct effect of competitive class climate and cyberloafing was not significant, and H5 was not supported, the two results need to be discussed.
A: Thank you for this reminder. While the study found that the direct effect of competitive class climate and cyberloafing was not significant, perceived stress plays a mediating role between competitive class climate and cyberloafing. On the one hand, this may be due to the fact that perceived stress may be "closer" to competitive class climate, and competitive climate may have an effect on other variables through perceived stress. This is similar to the findings of Fletcher et al. (2008), where competitive psychological climate was associated with greater stress but was not directly related to self-rated task performance. On the other hand, the intensity of the competitive classroom climate in today's society occurs within a " controlled range" and does not grow indefinitely, so it may not yet dampen other influences on adolescents' intrinsic psychological traits (Wang, Degol, Amemiya, Parr, & Guo, 2020); other factors that influence cyberloafing mainly affect adolescents' levels of perceived stress, which in turn affects their cyberloafing.
In addition, this study found that the moderating effect of self-esteem between competitive class climate and cyberloafing was not significant. This may be due to the fact that the effect of competitive class climate on cyberloafing occurs through perceived stress, and the direct relationship between them is not significant, which leads to the nonsignificant moderating effect of self-esteem between the two. According to the reviewer's comment, we have added the relevant discussion to the discussion section of the manuscript.
Fletcher, T. D., Major, D. A., & Davis, D. D. The interactive relationship of competitive climate and trait competitiveness with workplace attitudes, stress, and performance. Journal of Organizational Behavior: The International Journal of Industrial, Occupational and Organizational Psychology and Behavior. 2008,29(7), 899-922. https://doi.org/10.1002/job.503
Wang, M. T., Degol, J. L., Amemiya, J., Parr, A., & Guo, J. Classroom climate and children’s academic and psychological wellbeing: A systematic review and meta-analysis. Developmental Review. 2020, 57, 100912. https://doi.org/10.1016/j.dr.2020.100912
Reviewer 3 Report
Dear authors, thank you for an interesting article but can you please indicate in your abstract what cyberloafing is
Author Response
Dear authors, thank you for an interesting article but can you please indicate in your abstract what cyberloafing is?
A: Kamin (1995) first used the term "cyberloafing" to refer to the work deviant behavior of employees who surf the internet for entertainment during work (Kamin, 1995; Quinion, 2005). Later, different researchers named and defined this type of behavior differently. For example, Guthrie and Gray (1996) called it junk computing, which is the use of information systems in a way that is inconsistent with organizational goals, emphasizing its uselessness. Marron (2000) defined cyberloafing in terms of its harmfulness, stating that cyberloafing is "The behavior of employees who waste time and company resources by engaging in online entertainment during working hours". Mills and Hu ( 2001) define cyberloafing as the behavior of employees who use the internet for entertainment or personal purposes while at work. Lim ( 2002) considers cyberloafing to be the behavior of employees who spontaneously use the company network to browse non-work related websites and send and receive non-work related emails during work hours. Lim (2002)'s definition of cyberloafing has been widely cited and argued; therefore, we apply his view to the learning environment and define Cyberloafing as referring to an individual's spontaneous use of the internet to browse websites unrelated to work/study and send and receive emails or text messages unrelated to work/study while working/studying. Following your advice, we have added this definition of cyberloafing to the abstract.
Kamin L J. Behind the curve. Scientific American, 1995, 272(2), 99-103.
Lim, V. K. The IT way of loafing on the job: Cyberloafing, neutralizing and organizational justice. Journal of organizational behavior: the international journal of industrial, occupational and Organizational Psychology and Behavior. 2002, 23(5), 675-694. https://doi.org/10.1002/job.161
Marron,K. Attack of the cyberslackers. The ( London) Globe and Mail. 2000,January 20,p. T5.
Mills J E, Hu B, Beldona S, et al. Cyberslacking! A liability issue for wired workplaces. Cornell Hotel and Restaurant Administration Quarterly. 2001, 42(5), 34-47.https://doi.org/ 10.1016/S0010-8804(01)80056-2
Reviewer 4 Report
I have completed a review of the revised manuscript: "The relationship between competitive class climate and cyberloafing among Chinese adolescents: A curvilinear moderated mediation model".I want to thank the authors for clarifying all issues and updating the manuscript. My decision is "accept as is".Best regardsAuthor Response
I have completed a review of the revised manuscript: "The relationship between competitive class climate and cyberloafing among Chinese adolescents: A curvilinear moderated mediation model".I want to thank the authors for clarifying all issues and updating the manuscript. My decision is "accept as is".Best regards
A: Thank you for your affirmation of our work.
Reviewer 5 Report
Cyberloafing is a matter of learning ethics or research ethics, such as plagiarism. Thus, it will be helpful to find theories and examples that can be referred to in the field of pedagogy.
Author Response
Cyberloafing is a matter of learning ethics or research ethics, such as plagiarism. Thus, it will be helpful to find theories and examples that can be referred to in the field of pedagogy.
A: Thank you for this reminder. Based on the reviewer's comment, we added the relevant description to the Conclusion. For teachers, our results suggest that an appropriately competitive class climate is helpful in reducing individual cyberloafing, but it is also important to note that if the classroom climate is too competitive, it can increase the risk of adolescent cyberloafing. Therefore, teachers need to construct student-centered lesson management programs that make classroom management more participatory, allow students to set their own standards of behavior, and create a reasonably competitive environment that encourages students to achieve multiple successes (Freiberg & Lamb, 2009).
Freiberg, H. J., & Lamb, S. M. Dimensions of person-centered classroom management. Theory into practice. 2009, 48(2), 99-105. https://doi.org/10.1080/00405840902776228
Round 2
Reviewer 2 Report
The authors have responded well to the review comments, the manuscript can be accepted for publication. There are two small problems that need to be modified: first, the concept of Cyberloafing should not appear in the abstract; second, the appearance of the conceptual model (Figure 1. Hypothesized Model) is not appropriate enough, the lines should be drawn straight lines.
Author Response
The authors have responded well to the review comments, the manuscript can be accepted for publication. There are two small problems that need to be modified: first, the concept of Cyberloafing should not appear in the abstract; second, the appearance of the conceptual model (Figure 1. Hypothesized Model) is not appropriate enough, the lines should be drawn straight lines.
A: Thank you for recognizing our work. Based on the comments of the reviewers, we deleted the definition of cyberloafing from the abstract and redrew the conceptual model figure.